# Association of Clinical and Radiological Features with Disease Severity of Symptomatic Immune Checkpoint Inhibitor-Related Pneumonitis

**DOI:** 10.3390/diagnostics13040691

**Published:** 2023-02-12

**Authors:** Qian Zhang, Xiuli Tao, Shijun Zhao, Ning Li, Shuhang Wang, Ning Wu

**Affiliations:** 1Department of Diagnostic Radiology, National Cancer Center/National Clinical Research Center for Cancer/Cancer Hospital, Chinese Academy of Medical Sciences and Peking Union Medical College, Beijing 100021, China; 2Department of Nuclear Medicine (PET-CT Center), National Cancer Center/National Clinical Research Center for Cancer/Cancer Hospital, Chinese Academy of Medical Sciences and Peking Union Medical College, Beijing 100021, China; 3Department of Clinical Trial Center, National Cancer Center/National Clinical Research Center for Cancer/Cancer Hospital, Chinese Academy of Medical Sciences and Peking Union Medical College, Beijing 100021, China; 4Department of Diagnostic Radiology, National Cancer Center/National Clinical Research Center for Cancer/Hebei Cancer Hospital, Chinese Academy of Medical Sciences, Langfang 065001, China

**Keywords:** immune checkpoint inhibitors, checkpoint inhibitor-related pneumonitis, drug toxicity, computed tomography

## Abstract

Objectives: To investigate the predictive ability of clinical and chest computed tomography (CT) features to predict the severity of symptomatic immune checkpoint inhibitor-related pneumonitis (CIP). Methods: This study included 34 patients diagnosed with symptomatic CIP (grades 2–5) and divided into mild (grade 2) and severe CIP (grades 3–5) groups. The groups’ clinical and chest CT features were analyzed. Three manual scores (extent, image finding, and clinical symptom scores) were conducted to evaluate the diagnostic performance alone and in combination. Results: There were 20 cases of mild CIP and 14 cases of severe CIP. More severe CIP occurred within 3 months than after 3 months (11 vs. 3 cases, *p* = 0.038). Severe CIP was significantly associated with fever (*p* < 0.001) and the acute interstitial pneumonia/acute respiratory distress syndrome pattern (*p* = 0.001). The diagnostic performance of chest CT scores (extent score and image finding score) was better than that of clinical symptom score. The combination of the three scores demonstrated the best diagnostic value, with an area under the receiver operating characteristic curve of 0.948. Conclusions: The clinical and chest CT features have important application value in assessing the disease severity of symptomatic CIP. We recommend the routine use of chest CT in a comprehensive clinical evaluation.

## 1. Introduction

Recently, immune checkpoint inhibitors (ICIs) have emerged as a critical therapeutic approach for various malignancy types and have greatly improved clinical outcomes [1,2,3,4]. However, overactivation of the immune system leads to immune-related adverse events that can affect virtually any body organ [5,6,7]. Immune checkpoint inhibitor-related pneumonitis (CIP) has potentially fatal toxicity [7,8]. Clinically, the grade of CIP is classified according to the Common Terminology Criteria for Adverse Events (CTCAE) [9]. Grade 1 CIP cases are incidentally detected by computed tomography (CT) imaging with no symptoms, which presents relatively limited clinical significance because no further intervention is required. Therefore, we considerably pay attention to symptomatic CIP (grades 2–5), which is defined as the occurrence of new or aggravating respiratory symptoms such as dyspnea and cough, including new inflammatory lesions on chest CT imaging after ICI treatment and excluding pulmonary infection, tumor progression, and other reasons [10,11]. Symptomatic CIP leads to the delay and termination of ICI treatment [12]. Patients with severe disease may experience acute respiratory failure and treatment-related death [7,8], which is of great concern in clinical practice.

Even though CTCAE is the standard for evaluating CIP severity based on clinical features, guidelines have been updated to include radiological indicators as critical factors. The American Society of Clinical Oncology/National Comprehensive Cancer Network guideline [13] originally included the extent of pneumonitis on chest CT as a grading indicator, and the same approach was used in subsequent guidelines [14,15]. Corticosteroid therapy is the basic treatment for CIP, and the appropriate dose and duration of corticosteroids for mild (grade 2) and severe CIP (grades 3–5) are different. Insufficient corticosteroid administration worsens pneumonitis, whereas excessive corticosteroid use results in severe opportunistic infections. Therefore, the evaluation of CIP severity is essential for clinical decision-making. Notably, radiological features on chest CT are more intuitive and easier to quantify, which may be of great value.

This study aimed to retrospectively analyze the clinical features and chest CT imaging differences between mild and severe CIP. Additionally, we established three manual scores and examined the diagnostic performance of the scores alone and in combination.

## 2. Methods

### 2.1. Patient Selection

Patients who received immunotherapy for various malignancies in our hospital between January 2017 and April 2021 were retrospectively reviewed. CIP was defined as the development of new pulmonary inflammatory lesions after immunotherapy and was considered to be associated with immunotherapy by attending physicians. The following cases were excluded: (a) patients lacking chest CT scans before and during ICI treatment, (b) patients who underwent immunotherapy combined with thoracic radiotherapy, (c) patients exhibiting likelihood of active lung infection, and (d) patients exhibiting likelihood of cancer progression or malignant lung infiltration. The CIP grade was determined using the CTCAE version 5.0 [9]. This study focused on symptomatic CIP (grades 2–5) because asymptomatic CIP (grade 1) presented relatively limited clinical significance. The patients were categorized into mild (grade 2) and severe CIP (grades 3–5) groups. Notably, the onset time of CIP was defined as the time from the first use of ICIs to the occurrence of pneumonitis, and complete medical records were retrospectively collected accordingly.

### 2.2. Chest CT Examination

Serial chest CT images at CIP diagnosis were viewed by two radiologists (Q.Z. and X.L.T., with 3 and 16 years of experience in chest imaging, respectively). When the two radiologists had disagreements, a third radiologist (N.W., with 40 years of experience in chest imaging) reviewed the images independently and made the final assessment. We also consulted an expert in the field of pneumonitis, and disagreements were resolved by consensus. The radiologists had access to clinical details and previous images to ensure the correct differentiation of the target CIP region and to determine if patients had received previous radiotherapy, had lesions of cancer progression, or had any pre-existing lung abnormalities (such as chronic obstructive pulmonary disease or prior interstitial lung disease). Notably, the radiologists were blinded to the CIP grade when viewing the chest CT images.

Since all patients were required to undergo high-resolution CT, we obtained the standard clinical protocol from our hospital with ≥64-detector row scanners. The CT scanning protocols varied between unenhanced and enhanced scans because of their retrospective nature. Axial CT images were reconstructed with 1.25-mm or 5-mm thickness, and CT images were viewed at the lung window setting (width, 1500 HU, and level, −650 HU).

### 2.3. Evaluation of Chest CT Findings

Chest CT characteristics were assessed utilizing methods from previous treatment-related pneumonitis studies [16,17,18,19]. We evaluated CIP for the following features: (a) lung lobe involvement (right upper lobe, right middle lobe, right lower lobe, left upper lobe, and left lower lobe); (b) peripheral, central, mixed, or diffuse distribution in the field; (c) symmetrical or asymmetrical; (d) image findings, including ground-glass opacities (GGOs), consolidation, reticular opacities, interlobular septal thickening, honeycombing, and pleural effusion; and (e) a suggestive radiographic pattern based on the American Thoracic Society/European Respiratory Society international multidisciplinary classification of interstitial pneumonia [20,21,22] as follows: (i) organizing pneumonia (OP) pattern; (ii) nonspecific interstitial pneumonia (NSIP) pattern; (iii) hypersensitivity pneumonitis (HP) pattern; (iv) bronchiolitis pattern; (v) acute interstitial pneumonia (AIP)/acute respiratory distress syndrome (ARDS) pattern; and (vi) not applicable.

Furthermore, we designed three manual scores with the following semi-quantitative measurement parameters: (a) extent score: extent in the ratio of the volume in the upper (above the carina), middle (below the carina up to the inferior pulmonary vein), and lower (below the inferior pulmonary vein) lung zones (overall, six lung zones were assigned using a 6-point scale [0 point: none, 1 point: ≤5%, 2 points: 6–25%, 3 points: 26–50%, 4 points: 51–75%, and 5 points: 76–100% of lung parenchyma involved], and the total score ranged from 0 to 30 points); (b) image finding score: image findings, including GGO, consolidation, reticular opacities, interlobular septal thickening, honeycombing, and pleural effusion were assigned 1 point (yes) or 0 point (no), with the total score ranging from 0 to 6 points; and (c) clinical symptom score: clinical symptoms, including dyspnea, cough, wheezing, fever, chest tightness, chest pain, and hemoptysis, were assigned 1 point (yes) or 0 point (no), with the total score ranging from 0 to 7 points.

### 2.4. Statistical Analyses

Statistical tests were conducted using SPSS v. 25.0 (IBM Corp.). We descriptively analyzed the clinical and chest CT differences between patients with mild and severe CIP. Categorical data are presented as numbers (percentages), and continuous data are presented as medians (ranges). The differences between groups were evaluated using Fisher’s exact test for categorical variables and the Mann–Whitney U test for continuous variables. Subsequently, logistic regression analysis was used to create a combination score based on the extent, image finding, and clinical symptom scores. Receiver operating characteristics (ROC) were obtained to assess the performance of extent, image finding, clinical symptom, and the combination scores. For the combination score, ROC was constructed using the sum of the values obtained by weighing the parameters for the coefficients obtained by stepwise logistic regression analysis. Cutoff values were chosen based on optimal sensitivity. The area under the receiver operating characteristic curve (AUC), sensitivity, specificity, accuracy, positive predictive value, and negative predictive value were calculated. All *p*-values were based on a two-sided hypothesis, and *p* < 0.05 indicated statistical significance.

## 3. Results

### 3.1. Patient Characteristics and Clinical Features

Overall, 50 suspected CIP cases were observed in the 994 patients treated with immunotherapy. After review, 16 cases were excluded: infectious pneumonitis (n = 4), tumor progression (n = 2), aspiration pneumonitis (n = 1), radiation pneumonitis (n = 2), asymptomatic grade 1 CIP (n = 4), and lack of clinical or imaging data (n = 3). Finally, 34 patients with symptomatic CIP and available chest CT findings were enrolled. Figure 1 shows the flowchart of this study, and Table 1 lists the baseline characteristics and clinical features. Seventeen (50.0%) patients were treated with immunotherapy alone, including 13, 3, and 1 who received a programmed death-1 (PD-1) inhibitor, a programmed death ligand-1 inhibitor, and a herpes simplex virus type II coding for the granulocyte-macrophage colony-stimulating factor combined with a PD-1 inhibitor, respectively. Seventeen (50.0%) other patients were treated with combination therapy, including 13 who received a PD-1 inhibitor combined with chemotherapy, 1 who received a PD-1 inhibitor combined with antiangiogenesis, and 3 who received a PD-1 inhibitor combined with chemotherapy and antiangiogenesis.

Of the 34 patients included in this study (30 men and 4 women; median age, 60 years), 20, 10, 2, and 2 experienced grades 2, 3, 4, and 5 CIP, respectively. Mild CIP accounted for 58.8% (20/34) of the cases, while severe CIP accounted for 41.2% (14/34). Cough (n = 29, 85.3%) and dyspnea (n = 27, 79.4%) were the most common presenting symptoms, while fever was associated with severe CIP (*p* < 0.001). Following immunotherapy, the median onset time of CIP was 77.5 (range; 10–228) days, and CIP occurred within 3 months in 55.9% (19/34) patients. Of the 14 patients with severe CIP, more severe CIP cases occurred within 3 months than after 3 months (11 vs. 3 cases, *p* = 0.038).

### 3.2. Radiological Features

Table 2 summarizes the CT characteristics according to the CIP grade. Six patients had a history of pulmonary lobectomy: four with one lobe excised, one with two lobes excised, and one with wedge resection of two lobes. The median number of lung lobes involved was 3 (range, 1–5). Patients with severe CIP had more lung lobes involved than those with mild CIP (4 vs. 2 lobes, *p* = 0.010). Pneumonitis on chest CT showed an asymmetrical distribution in 85.3% (29/34) patients. GGOs (n = 33, 97.1%) and consolidation (n = 29, 85.3%) were the most commonly observed CT image findings, and the OP pattern (n = 15, 44.1%) was the most common pattern.

CT findings associated with severe CIP included increased lung lobe involvement (*p* = 0.010), diffuse distribution (*p* = 0.005), reticular opacities (*p* = 0.038), interlobular septal thickening (*p* = 0.001), honeycombing (*p* = 0.022), pleural effusion (*p* = 0.042), and an AIP/ARDS pattern (*p* = 0.001). In contrast, those associated with mild CIP included a peripheral distribution (*p* = 0.024), an HP pattern (*p* = 0.031), and an OP pattern (*p* = 0.038).

### 3.3. Diagnostic Performance of Three Manual Scores and the Combination Score for Severe CIP

Table 3 shows the comparison of diagnostic performance to differentiate between mild and severe CIP, and Figure 2 illustrates the ROC curves. Our results indicated that the extent score (AUC 0.857) and image finding score (AUC 0.843) presented better performances than the clinical symptom score (AUC 0.782). The combination of the three manual scores (−10.562 + 0.132 × Extent score + 0.748 × Image finding score + 2.154 × Clinical symptom score) demonstrated the best diagnostic value over the three scores alone, with an AUC of 0.948. Additionally, Figure 3 shows the CT images of three representative patients with mild, severe, and fatal CIP.

## 4. Discussion

Severe CIP has potentially fatal toxicity. Challenges occur in severe CIP in clinical decision-making, including the appropriate dose and duration of corticosteroids and ICI rechallenge. It is crucial to perform risk stratification of CIP patients in the early stages to determine the severity and enable timely diagnosis and treatment. Our study aimed to analyze the clinical and chest CT imaging differences between mild and severe CIP. Additionally, we established three manual scores and examined the diagnostic performance of the three scores alone and in combination when assessing CIP severity. Our results revealed that more severe CIP occurred within 3 months than after 3 months. Severe CIP was significantly associated with fever and the AIP/ARDS pattern. The diagnostic performance of chest CT scores (extent score and image finding score) was better than that of clinical symptom score. The combination of chest CT features and clinical symptoms demonstrated the best diagnostic performance. The findings of this study demonstrated the importance of chest CT, which plays a significant role in clinical comprehensive assessment.

CIP is a diagnosis of exclusion based on consensus experience [13,14,15]. Proof of drug administration, temporal eligibility (symptom development following drug initiation), and an appropriate latency period between drug administration and the development of symptoms help raise suspicion of CIP. Diagnostic confirmation requires the exclusion of infection, tumor progression, carcinomatous lymphangitis, radiation-related pneumonitis, thromboembolism, and pulmonary edema. Diagnostic confirmation also requires a comprehensive consideration of clinical symptoms, chest CT imaging, and tests of sputum, blood, urine cultures, and a nasal swab. In general, routine lung biopsies are not recommended. However, if there is clinical or radiological doubt about the etiology of pulmonary infiltrates, a lung biopsy may provide an answer.

This study observed a positive correlation between CIP severity and the extent, image finding, and clinical symptom scores. First, the extent could indicate the severity of CIP, and an extent score over 11 points meant lesion deterioration in our study. Clinical guidelines indicated that the extent of lung parenchyma involvement was <50% for mild CIP and >50% for severe CIP [13,14,15]. The extent also reflected the severity of radiation pneumonitis, viral pneumonia, and other drug-associated pneumonitis [23,24,25]. Second, the image finding score quantified CT findings. It could be speculated that as the disease course deteriorated with a higher score, CT findings could show more diverse and complex pulmonary opacities, in addition to GGOs and consolidation. Third, clinical symptoms are subjective and susceptible to patient status, age, tumor stage, and previous pulmonary disease. Our study found that the clinical symptom score had the lowest sensitivity (0.429) and that the diagnostic performance of chest CT scores was better than that of the clinical symptom score. Therefore, the assessment of the chest CT might be more important when patients had mild symptoms in the early stage. Fourth, the combination of the three scores demonstrated the best diagnostic value over the three scores alone, indicating that the comprehensive evaluation of chest CT and clinical symptoms is of great value in clinical application. Finally, a chest CT could help monitor disease progression during the follow-up period. The improvement of CIP manifests as a total or partial resolution of pneumonitis and a decreased score. In contrast, CIP progresses to a more severe phase with an increased score.

This study revealed that more severe CIP occurred within 3 months than after 3 months (11 vs. 3 cases, *p* = 0.038). This finding is in line with that of the study conducted by Huang et al. [26], with 6 weeks as the dividing line (92.9% vs. 7.1%, *p* < 0.05). Huang et al. reported that early-onset CIP had higher radiologic severity and a poorer prognosis, with an OP pattern as the dominant radiographic pattern, while late-onset CIP had lower radiologic severity and a better prognosis, with an NSIP pattern as the dominant radiographic pattern [26]. We speculated that a possible explanation was that overactivation of the immune system in early-onset CIP patients might cause more violent inflammatory cytokine cascade responses or even undergo a cytokine storm in severe and fatal cases. However, a study by Delaunay et al. [16] pointed out that there appeared to be no correlation between the occurrence time and clinical severity (*p* = 0.32). The possibility of an early onset of severe CIP requires further verification. Nevertheless, it demonstrated the clinical significance that physicians should perform necessary follow-ups to ensure the detection of severe CIP in the early stages, especially within 3 months. Our study indicated that fever was a significant predictor of severe CIP (*p* < 0.001). Although the relationship between fever and CIP severity is poorly understood, we speculated that patients with fever could cause the spread of inflammatory factors throughout the body, which might indicate a more severe systemic response than localized respiratory symptoms. In clinical practice, fever also plays a role in distinguishing CIP from infectious pneumonia. CIP was less prone to fever and more prone to cough and dyspnea, and fever tended to be mild to moderate if it occurred [12,16,17,19,27,28].

Radiological features could reflect CIP severity to a certain extent. We observed that severe CIP was associated with increased lung lobe involvement, diffuse distribution, reticular opacities, interlobular septal thickening, honeycombing, and pleural effusion. Previous reports on severe CIP cases demonstrated similar CT characteristics of extensive and diffuse findings in both lungs [17,29,30]. These results indicated that as the severity of CIP increased, the effect of pneumonitis evolved from localized lung injury to a wider inflammatory response and showed complex pulmonary opacities. The AIP/ARDS pattern has been associated with the most severe clinical course and was the risk factor for CIP-related deaths [17,22]. The AIP/ARDS pattern was a significant predictor of severe CIP (*p* = 0.001) in our study. All seven patients with the AIP/ARDS pattern had severe CIP, and two died during follow-up. This finding was correlated with the pathological characteristics of severe and fatal CIP, and it indicated the extension and deterioration of the disease course. A pathological study revealed that two patients with fatal CIP had total lung injury [31]: one showed diffuse alveolar damage with foamy macrophage accumulation and eosinophilic hyaline membranes and that the other patient showed acute fibrinous pneumonitis with alveolar septal edema, abundant foamy macrophages in the airspaces, intra-alveolar fibrin, and reactive pneumocyte hyperplasia with vacuolization. Previous studies reported that the OP pattern (some studies called it the COP pattern) was the most frequent radiographic pattern of CIP [12,16,17,19,27,28], which is consistent with our study. Nishino et al. [17] also reported that the OP pattern was the most common in different tumor types and in both monotherapy and combination therapy. Notably, this finding is similar to other drug-related and radiation pneumonitis [23,24,32]. These results might indicate that pneumonitis was a general manifestation of the lung’s response to various injuries, and the OP pattern was the most common form.

This study has some limitations. First, this was a single-center retrospective study with a relatively small sample size; hence, further studies with larger sample sizes are warranted for verification. Second, some patients received combination therapy, such as chemotherapy and antiangiogenesis, which might have affected pneumonitis from additional agents. Tumor patients, particularly advanced tumor patients after multiline treatment, had complex clinical courses. However, our retrospective study might be more closely related to actual clinical situations than a prospective cohort with ICI monotherapy. Third, subjective assessment of radiological features could lead to interobserver bias, and we attempted to reduce it by reaching consensus contours among three radiologists and consulting an expert. Finally, the segmentation of the lesion volume on CT was based on visual and semi-quantitative measurements. Remarkably, accurate and quantitative segmentation using artificial intelligence software is a future research direction.

## 5. Conclusions

More severe CIP occurred within 3 months than after 3 months. Severe CIP was significantly associated with fever and the AIP/ARDS pattern. The diagnostic performance of chest CT scores was better than that of clinical symptom score in assessing the CIP severity. The combination of chest CT features and clinical symptoms had the best diagnostic performance. The findings of this study demonstrated the importance of chest CT, which plays a significant role in clinical comprehensive assessment. Based on the evidence of the current study, we therefore recommend that a chest CT be routinely used in a comprehensive clinical evaluation.

## Figures and Tables

**Figure 1 diagnostics-13-00691-f001:**
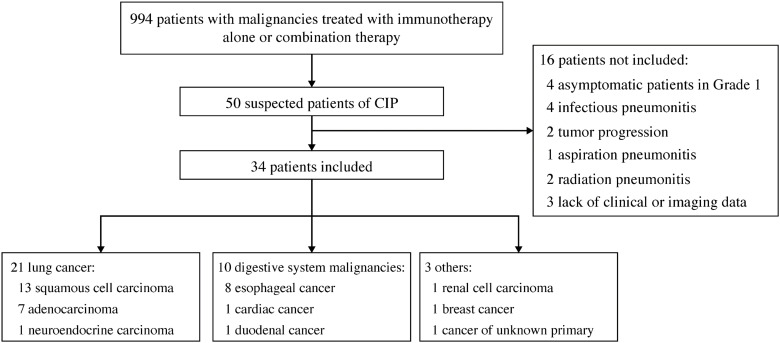
Study flowchart. Abbreviation: CIP, checkpoint inhibitor-related pneumonitis.

**Figure 2 diagnostics-13-00691-f002:**
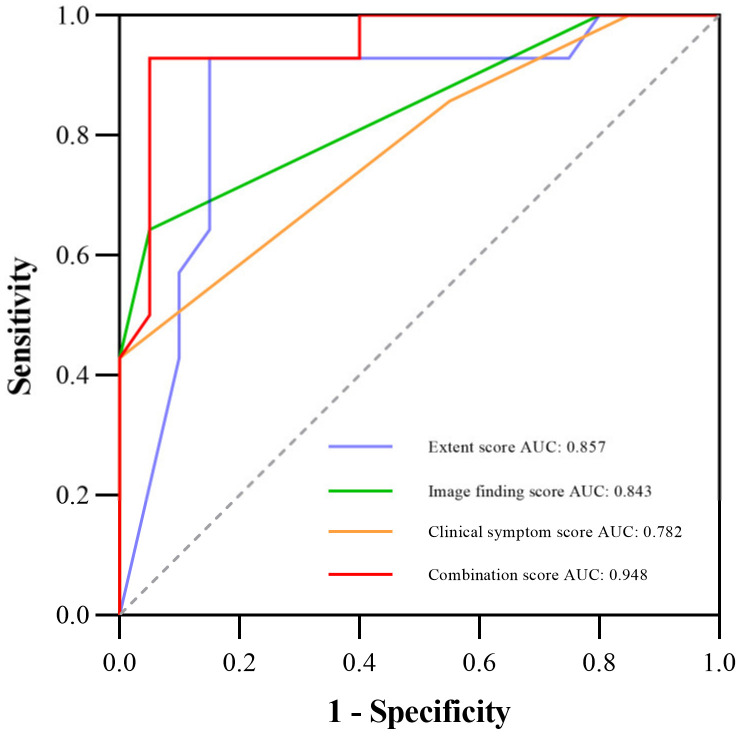
ROC curve for diagnostic performance among the extent, image finding, clinical symptom, and the combination scores; the AUCs are 0.857, 0.843, 0.782, and 0.948, respectively. Abbreviations: AUC, the area under the receiver operating characteristic curve; ROC, the receiver operating characteristic.

**Figure 3 diagnostics-13-00691-f003:**
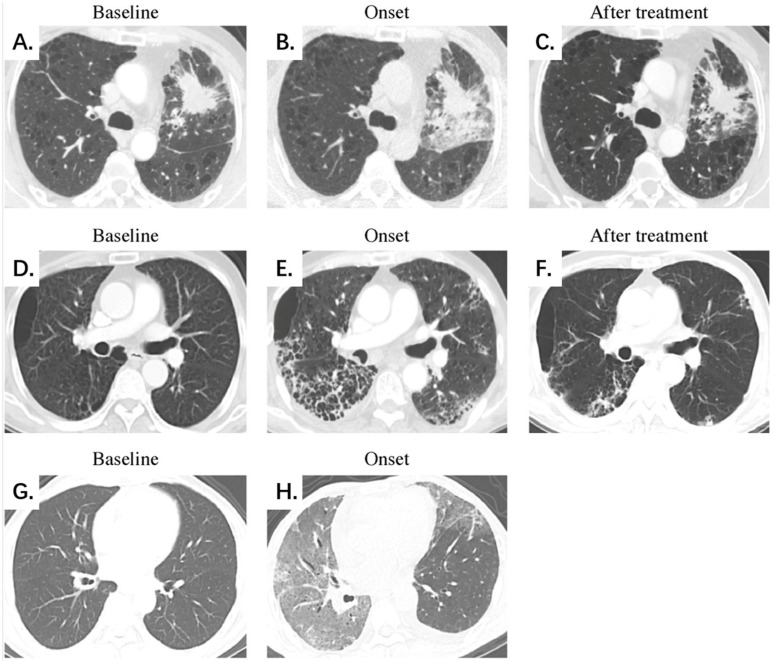
Representative chest CT scans of mild, severe, and fatal CIP at different stages. (**A**–**C**) Mild CIP with an OP pattern. A 51-year-old male patient with lung adenocarcinoma was treated with a PD-1 inhibitor combined with chemotherapy. Chest CT on day 24 after ICI initiation demonstrates localized development of GGOs, consolidation, and reticular opacities mainly surrounding the tumor lesion. The patient presented symptoms of dyspnea, cough, and wheezing. The extent, image finding, and clinical symptom scores were 6, 3, and 3 points for the whole lung, respectively, and CIP demonstrates significant absorption after oral corticosteroid treatment. (**D**–**F**) Severe CIP with an NSIP pattern. A 64-year-old male patient with squamous cell lung carcinoma was treated with a PD-1 inhibitor combined with chemotherapy. Chest CT on day 38 after ICI initiation demonstrates a diffuse development of GGOs, consolidation, reticular opacities, interlobular septal thickening, honeycombing, and a small amount of pleural effusion dominated subpleural distribution. The patient presented symptoms of dyspnea, wheezing, fever, and hemoptysis. The extent, image finding, and clinical symptom scores were 18, 6, and 4 points for the whole lung, respectively. CIP shows significant absorption after intravenous corticosteroid treatment. (**G**,**H**) Fatal CIP with an AIP/ARDS pattern. A 60-year-old male patient with duodenal cancer was treated with a herpes simplex virus type II coding for the granulocyte-macrophage colony-stimulating factor combined with a PD-1 inhibitor. Chest CT on day 78 after ICI initiation demonstrates a diffuse development of GGOs, consolidation, reticular opacities, interlobular septal thickening, and a small amount of pleural effusion, presenting the unusual manifestation of the “crazy paving” sign. The patient presented symptoms of dyspnea, cough, and hemoptysis. The extent, image finding, and clinical symptom scores were 30, 5, and 3 points for the whole lung, respectively. The patient died early of respiratory failure, and no follow-up CT scan was available. Abbreviations: CIP, checkpoint inhibitor-related pneumonitis; PD-1, programmed cell death-1; ICIs, immune checkpoint inhibitors; GGO, ground-glass opacity; OP, organizing pneumonia; NSIP, nonspecific interstitial pneumonia; AIP/ARDS, acute interstitial pneumonia/acute respiratory distress syndrome.

**Table 1 diagnostics-13-00691-t001:** Patient characteristics by grades.

	All Patients	Mild CIP	Severe CIP	*p* Value
Tumor type				0.477
Lung cancer	21	11 (52.4%)	10 (47.6%)	
Non-lung cancer	13	9 (69.2%)	4 (30.8%)	
Sex				0.627
Female	4	3 (75.0%)	1 (25.0%)	
Male	30	17 (56.7%)	13 (43.3%)	
Age, y	60 (38–77)	62 (38–77)	58 (52–70)	0.713
ECOG				1.000
0	11	7 (63.6%)	4 (36.4%)	
1	21	12 (57.1%)	9 (42.9%)	
2	2	1 (50.0%)	1 (50.0)	
Smoking status				0.880
Never smoker	8	5(62.5%)	3 (37.5%)	
Former smoker	22	12 (54.5%)	10 (45.5%)	
Current smoker	4	3 (75.0%)	1 (25.0%)	
Pack-years	40 (4–100)	40 (4–80)	30 (20–100)	0.979
Family history of malignancy				0.477
Yes	13	9 (69.2%)	4 (30.8%)	
No	21	11 (52.4%)	10 (47.6%)	
History of fibrosis and emphysema				1.000
Yes	16	9 (56.3%)	7 (43.8%)	
No	18	11 (61.1%)	7 (38.9%)	
History of chest radiation therapy				0.092
Yes	16	12(75.0%)	4 (25.0%)	
No	18	8 (44.4%)	10 (55.6%)	
History of pulmonary lobectomy				0.202
Yes	6	2 (33.3%)	4 (66.7%)	
No	28	18 (64.3%)	10 (35.7%)	
Regimen of immune therapy				1.000
Immunotherapy alone	17	10 (58.8%)	7 (41.2%)	
Combination therapy	17	10 (58.8%)	7 (41.2%)	
Symptoms				
Dyspnea	27	16 (59.3%)	11 (40.7%)	1.000
Cough	29	17 (58.6%)	12 (41.4%)	1.000
Wheezing	19	13 (68.4%)	6 (31.6%)	0.296
Fever	8	0 (0.0%)	8 (100.0%)	<0.001
Chest tightness	7	2 (28.6%)	5 (71.4%)	0.097
Hemoptysis	3	0 (0.0%)	3 (100.0%)	0.061
Chest pain	1	0 (0.0%)	1 (100.0%)	0.412

Abbreviations: CIP, checkpoint inhibitor-related pneumonitis; ECOG PS, Eastern Cooperative Oncology Group Performance Status.

**Table 2 diagnostics-13-00691-t002:** Radiological features by grades.

	All Patients	Mild CIP	Severe CIP	*p* Value
Lung lobe involved	3 (1–5)	2 (1–5)	4 (1–5)	0.010
Distribution				
Peripheral distribution	10	9 (90.0%)	1 (10.0%)	0.024
Central distribution	2	2 (100.0%)	0 (0.0%)	0.501
Mixed distribution	8	5 (62.5%)	3 (37.5%)	1.000
Diffuse distribution	14	4 (28.6%)	10 (71.4%)	0.005
Image findings				
Ground-glass opacities	33	19 (57.6%)	14 (42.4%)	1.000
Consolidation	29	16 (55.2%)	13(44.8%)	0.379
Reticular opacities	19	8 (42.1%)	11(57.9%)	0.038
Interlobular septal thickening	12	2 (16.7%)	10(83.3%)	0.001
Honeycombing	4	0 (0.0%)	4 (100.0%)	0.022
Pleural effusion	8	2 (25.0%)	6 (75.0%)	0.042
Radiographic pattern				
OP pattern	15	12 (80.0%)	3 (20.0%)	0.038
NSIP pattern	3	1 (33.3%)	2 (66.7%)	0.555
HP pattern	6	6 (100.0%)	0 (0.0%)	0.031
Bronchiolitis pattern	1	1 (100.0%)	0 (0.0%)	1.000
AIP/ARDS pattern	7	0 (0.0%)	7 (100.0%)	0.001
Not applicable	2	0 (0.0%)	2 (100.0%)	0.162

Abbreviations: CIP, checkpoint inhibitor-related pneumonitis; OP, organizing pneumonia; NSIP, nonspecific interstitial pneumonia; HP, hypersensitivity pneumonitis; AIP/ARDS, acute interstitial pneumonia/acute respiratory distress syndrome.

**Table 3 diagnostics-13-00691-t003:** Comparison of diagnostic performance among extent, image finding, clinical symptom, and the combination scores.

	Cut-Off Value	AUC	95% (CI)	Sen.	Spe.	Acc.	PPV	NPV	*p* Value
Extent score	11	0.857	0.716–0.998	0.929	0.850	0.882	0.813	0.944	<0.001
Image finding score	4	0.843	0.702–0.984	0.643	0.950	0.824	0.900	0.792	<0.001
Clinical symptom score	4	0.782	0.622–0.942	0.429	1.000	0.765	1.000	0.714	0.006
Combination score *	0	0.948	0.873–1.000	0.929	0.950	0.912	0.923	0.905	<0.001

* Combination score = −10.562 + 0.132 × Extent score + 0.748 × Image finding score + 2.154 × Clinical symptom score. Abbreviations: CIP, checkpoint inhibitor-related pneumonitis; AUC, the area under the receiver operating characteristic curve; CI, confidence interval; Sen, sensitivity; Spe, specificity; Acc, accuracy; PPV, positive predictive value; NPV, negative predictive value.

## Data Availability

Data is contained within the article. Data supporting the reported results may be provided upon reasonable request.

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
