# Peer review of "Association of Clinical and Radiological Features with Disease Severity of Symptomatic Immune Checkpoint Inhibitor-Related Pneumonitis"

_diagnostics, 2023, doi:10.3390/diagnostics13040691_

Round 1
Reviewer 1 Report
Even if the study integrated the evaluation of a small group of patients, it was still a group that could develop and summarize clear conclusions about a pathology that needs clear protocols for diagnosis to allow targeted treatment. Neoplastic pathology confers the disadvantage of complications that do not necessarily have the same course compared to the general population.
Author Response
We would like to express our sincere gratitude for your careful assessment of our manuscript.
Please see the attachment for response letter.

Reviewer 2 Report
The manuscript is well written.
The main objective of the study was explained. I suggest a better explanation about the more severe CIP occurred within 3 months than after 3 months.
The discussion is in accordance with the results and the conclusion as well too
Author Response

(The authors gave the same response as above.)
